# Dopamine signaling tunes spatial pattern selectivity in *C. elegans*

**Bicheng Han†, Yongming Dong, Lin Zhang, Yan Liu, Ithai Rabinowitch, Jihong Bai***

Basic Sciences Division, Fred Hutchinson Cancer Research Center, Seattle, United States

**Abstract** Animals with complex brains can discriminate the spatial arrangement of physical features in the environment. It is unknown whether such sensitivity to spatial patterns can be accomplished in simpler nervous *systems* that lack long-range sensory modalities such as vision and hearing. Here we show that the nematode *Caenorhabditis elegans* can discriminate spatial patterns in its surroundings, despite having a nervous system of only 302 neurons. This spatial pattern selectivity requires touch-dependent dopamine signaling, including the mechanosensory TRP-4 channel in dopaminergic neurons and the D2-like dopamine receptor DOP-3. We find that spatial pattern selectivity varies significantly among *C. elegans* wild isolates. Electrophysiological recordings show that natural variations in TRP-4 reduce the mechanosensitivity of dopaminergic neurons. Polymorphic substitutions in either TRP-4 or DOP-3 alter the selectivity of spatial patterns. Together, these results demonstrate an ancestral role for dopamine signaling in tuning spatial pattern preferences in a simple nervous system.

*For correspondence: jbai@ fredhutch.org

Present address: †Department of Organismic and Evolutionary Biology, Center for Brain Science, Harvard University, Cambridge, United States

Competing interests: The authors declare that no competing interests exist.

## Introduction

Animals must relentlessly explore their surroundings in search of food, water, and shelter, in order to survive. Animals with complex nervous systems utilize multiple sensory modalities such as vision, touch, and hearing to perceive spatial patterns that reflect the physical structure of their environment. But can animals with much simpler nervous systems detect and discriminate spatial patterns in their environment as well? To address this question, we investigated the roundworm *C. elegans,* which feeds and reproduces in decomposing fruits and plant stems (*Barrière and Félix, 2014*). *C. elegans* has only 302 neurons and lack vision and hearing, mainly relying on smell and touch to sense their environment. *C. elegans* uses the sense of touch (or mechanosensation) to detect physical contact with external objects. For example, touch to the worm body or nose elicits reflex withdrawal responses (*Chalfie and Sulston, 1981*, *Chalfie and Wolinsky, 1990*; *Kaplan and Horvitz, 1993*; *Sulston et al., 1975*; *Way and Chalfie, 1989*), which allows the worms to quickly avoid potential danger. Beyond reflex, mechanosensation could also provide important information about the physical structure of the environment, aiding in navigation and spatial orientation. Indeed, mechanosensory detection of food elicits a slowing in *C. elegans* locomotion, increasing the time spent where food is present (*Sawin et al., 2000*). However, it remains unknown whether *C. elegans* mechanosensation is limited to such binary distinctions between the presence and absence of food, or whether it enables more general and refined perception of particular spatial patterns, and if so, how this depends on context and experience. These questions are important for determining to what extent a simple animal can sense the spatial arrangement of physical features in its surroundings, providing general insights about how animals perceive spatial patterns.

Here, we report that *C. elegans* has the ability to discriminate spatial patterns using mechanosensory-dependent dopamine signaling. This spatial pattern selectivity is not a fixed behavior. Instead, it is modified by experience and by internal states such as starvation. Among *C. elegans* wild

isolates, the absolute degree of spatial pattern preference is tuned by natural variations in the mechanosensory channel TRP-4 and the dopamine receptor DOP-3, leading to differential behavioral responses to identical environmental settings. These results suggest that the sense of the physical surroundings may undergo rapid evolutionary expansion to diversify the choice of preferred habitat, and perhaps to benefit the survival of the species.

## Results

To study how spatial patterns in its surroundings may impact worm behavior, we examined worm locomotion in microfluidic chambers comprising four patterns of PDMS (polydimethylsiloxane) pillars, designated quad-chambers (*Figure 1A* and *Figure 1—figure supplement 1A*). Pillar density increased in the four patterns from zero pillars (Pattern I) to the highest density of pillars (Pattern IV). We found that the Bristol N2 strain exhibits a strong preference for pattern IV (*Figure 1B*). Approximately 63% of N2 worms were found in the section containing pattern IV 60 min after entering the chamber. By contrast, we observed only 9–15% of worms in each of the other sections I, II and III (*Figure 1C*) with lower pillar densities. The strong preference for pattern IV was also indicated by a high positive value of a spatial pattern preference index (*Figure 1C*; Methods). To investigate the robustness of this pattern selectivity, we quantified N2 preference for pattern IV at various time points (*Figure 1—figure supplement 1B*). Pattern IV preference was very stable between 60 and 90 min, but was reduced at 120 min. This decrease was likely due to starvation, as addition of food (OP50 bacteria) into the chamber prolonged the preference for pattern IV, which lasted at least 180 min (*Figure 1—figure supplement 1C*). Consistent with this notion, worms that had already been starved prior to the experiment exhibited a significant reduction in pattern IV preference (*Figure 1—figure supplement 1D*), indicating that feeding state (starvation) can override an inherent preference for pattern selectivity.

To test whether pattern selectivity is based on individual choice or group behavior, we performed cumulative single-worm analyses. Single-worm pattern preference index values were similar to those obtained in population experiments (*Figure 1—figure supplement 2A*), indicating that pattern selectivity is an individual rather than group behavior. To identify the sensory modality used for pattern selection, we analyzed mutant worms that have defects in either oxygen or touch sensation. *C. elegans* exhibits aggregation behavior at optimal oxygen levels (*Cheung et al., 2004*; *Gray et al., 2004*). We thus examined whether oxygen sensation might play a role in the observed clustering within pattern IV. To this end, we tested mutant worms that lack the GCY-35 soluble guanylate cyclase and are defective in oxygen-dependent aggregation, bordering, and social feeding behaviors (*Cheung et al., 2004*; *Gray et al., 2004*). However, these mutant *gcy-35* worms continued to show preference for pattern IV (*Figure 1D*), indicating that pattern IV selectivity was independent of possible variations in oxygen levels and oxygen sensing. By contrast, mutant worms with dysfunctional MEC-10 DEG/ENaC channels showed a significant reduction in their preference for pattern IV (*Figure 1E*). To understand where MEC-10 functions, we performed cell-specific rescue experiments. As expected, single-copy *Pmec-10::mec-10* transgenes that drive MEC-10 expression in the six touch receptor neurons and in PVD and FLP neurons fully restored the preference for pattern IV. Expression of MEC-10 under *Pegl-44* in FLP and HSN neurons (*Wu et al., 2001*) also rescued the defects in *mec-10(tm1552)* mutant worms. By contrast, MEC-10 expression in the six touch receptor neurons (driven by *Pmec-18*) did not recover the preference for pattern IV. Together, these data indicate that MEC-10 acts in the polymodal nociceptive FLP neurons to support the selection of pattern IV.

To determine whether pattern selectivity depends on the particular absolute pillar density of pattern IV, we used a chamber consisting of patterns I and III only (*Figure 1—figure supplement 2D*). In this chamber, N2 worms showed a clear preference for pattern III (*Figure 1—figure supplement 2E*), even though this is a non-favored pattern in the quad-chamber. We therefore conclude that pattern selectivity is based on comparison between different spatial patterns *i.e.*, environmental context, rather than on the specific structural arrangement of pattern IV.

We next investigated the strategies that worms use to select pattern IV in the quad-chamber. First, we considered local effects occurring at the boundaries between adjacent patterns. We tracked worms that left pattern IV and entered into the less desired pattern III (*Figure 2A*), and found that these worms re-crossed back into the preferred pattern IV at a much higher rate than worms already within pattern IV (*Figure 2B*). We also found an increase in both reversal and turning

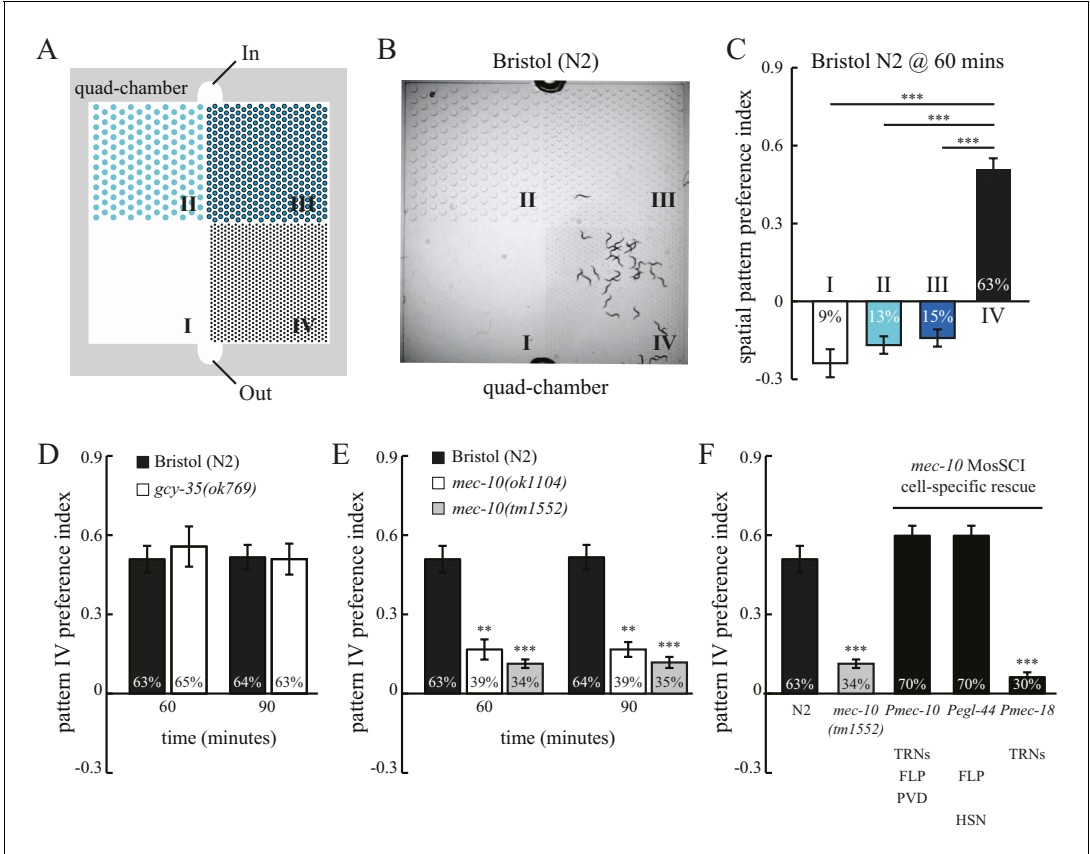

**Figure 1.** *C. elegans* exhibits spatial pattern selectivity. (**A**) A schematic drawing of the polydimethylsiloxane (PDMS) chamber used in the spatial pattern preference assay. Each chamber is divided into four sections (I–IV). The PDMS chamber (designated quad-chamber) is filled with M9 buffer. Worms are loaded with M9 buffer into the chamber through the 'In' port. Detailed information about the chamber dimensions is shown in *Figure 1—figure supplement 1A*. (**B**) *C. elegans* N2 strain prefers to remain in pattern IV that has the highest density of PDMS pillars. The image was taken 60 min after the worms were loaded into the chamber. (**C**) The degree of pattern preference was quantified at 60 min in two ways. First, the percentage of worms within each pattern was calculated (shown inside the bars). Second, spatial pattern preference index values for each section were obtained using the following formula: [# of worms in that section – (# of worms in the other three sections/3)] ÷ total # of worms. If worms are equally distributed among the different patterns, then all the index values are zero. If no worm is found in a given pattern, then the index value of that pattern is −0.33. If all worms are found within a single pattern, then the index value of that pattern is 1. n = 12; ***p<0.001 when compared to the index value of pattern IV; Dunnett's multiple comparison tests. (**D**) N2 worms and *gcy-35(ok769)* mutants show a similar preference for pattern IV. n = 6 for *gcy-35* mutants, and n = 10 for N2. (**E**) Pattern selectivity was significantly impaired in *mec-10(ok1104)* and *mec-10(tm1552)* mutant worms; n = 5 for each *mec-10* mutant, and n = 10 for N2. **p<0.01 and ***p<0.001 when compared with N2; Student's *t* test. (**F**) Cell-specific rescue of *mec-10(tm1552)* mutant worms. Transgenic worms that carry single-copy transgenes (MosSCI; *Pmec-10::mec-10cDNA*, *Pegl-44::mec-10cDNA*, or *Pmec-18::mec-10cDNA*) were examined for their pattern IV selectivity. n = 5 for each transgenic worm strain. ***p<0.001 when compared with N2; Dunnett's multiple comparison tests. Error bars denote s.e.m.

The following figure supplements are available for figure 1:

**Figure supplement 1.** Spatial pattern selectivity is modulated by feeding state.

**Figure supplement 2.** Spatial pattern selectivity is an individual behavior that relies on mechanosensation and spatial context.

rates when worms cross the IV-to-III boundary (*Figure 2C*). Therefore, worms sense local changes in pattern density, and respond quickly to unfavorable changes by reorientation, reinforcing our finding of their ability to make decisions based on comparisons of spatial density. Second, we examined whether worm locomotion within pattern IV was affected by the presence of neighboring patterns. We hypothesized that preceding experience of alternative spatial configurations would have a lasting impact on worm behavior within preferred pattern IV. To test this hypothesis, we designed an

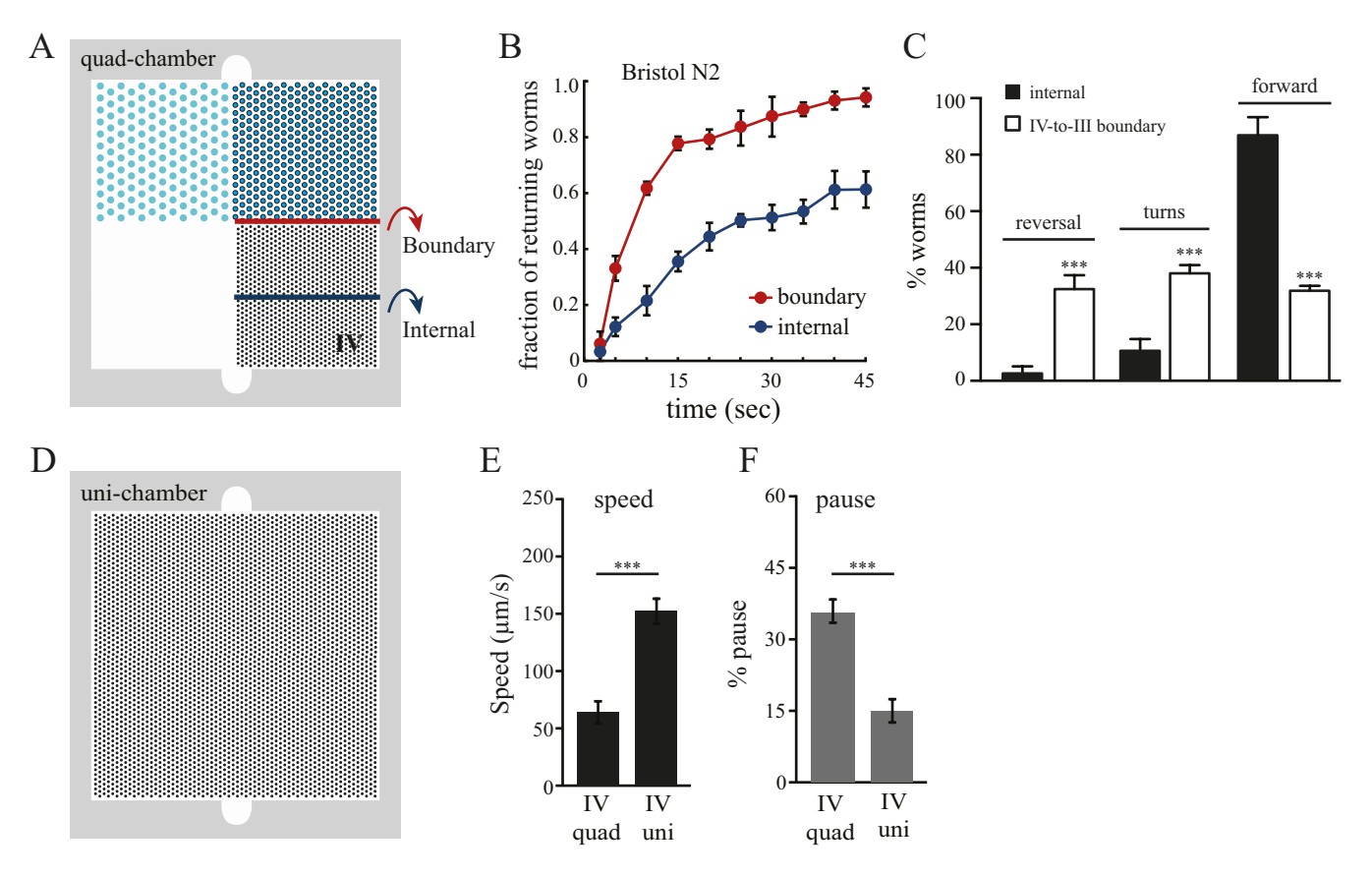

**Figure 2.** Boundary experience impacts *C. elegans* spatial pattern selectivity. Crossing the boundary from pattern IV to pattern III stimulates worms to return to pattern IV. (A) The red line indicates the boundary between patterns IV and III, and the blue line serves as a reference line in the middle of pattern IV. (B) The fraction of worms returning to pattern IV at indicated time points. Time zero indicates the moment that worms cross the red line (from pattern IV to III) or the blue line. Data were collected between 55–65 min after the worms were loaded into the chambers. n = 6. (C) Worms exhibit increased reversal and turning rates when they cross the pattern IV-to-III boundary. n = 6. (D) The design of the uniform chamber (uni-chamber). This chamber has only one pillar array pattern, which is identical to pattern IV in the quad-chamber. (E–F) Worms show different locomotion properties within pattern IV depending on the presence of absence of alternative surrounding patterns. The speed of moving (µm/sec) and the pause rates (%) were quantified from worms within pattern IV in either the quad-chamber or the uni-chamber. The number of independent repeats, n = 6; ***p<0.001; Student's *t* test. Error bars denote s.e.m.

additional chamber (designated uni-chamber; *Figure 2D*) containing only a pattern IV pillar configuration throughout its surface and no other patterns. We compared the speed and pause rates of worms within pattern IV in both quad- and uni- chambers. We found that worms in pattern IV of the quad-chamber moved more slowly and paused more than worms in the uni-chamber (*Figure 2E–F*). Such slowing and pausing could prolong the stay of worms in pattern IV. Together, these data suggest that the clustering of worms in pattern IV is due to both real-time responses to pattern changes upon boundary crossing and to longer-lasting effects caused by the experience of other less desired patterns. Thus, without vision or hearing, *C. elegans* can take advantage of mechanosensory information to compare and select between spatial patterns in its environment.

We next set to explore the molecular basis for the observed spatial pattern selectivity. Several features of this behavior prompted us to investigate the contribution of dopamine signaling. In *C. elegans*, dopamine regulates a variety of behaviors including locomotion, mechanosensation, food availability, touch habituation, and the swim-to-crawl transition (*Calhoun et al., 2015*; *Chase et al., 2004*; *Hills et al., 2004*; *Kindt et al., 2007*; *Sanyal et al., 2004*; *Sawin et al., 2000*; *Vidal-Gadea et al., 2011*). In particular, dopamine signals decrease locomotion rates when worms enter a food source (a lawn of bacteria) (*Sawin et al., 2000*), and increase returning frequency when worms

leave the bacteria lawn (*Hills et al., 2004*). To test whether dopamine is required also for spatial pattern selectivity, we examined *cat-2(e1112)* animals, which lack the CAT-2 tyrosine hydroxylase and are defective in dopamine biosynthesis (*Lints and Emmons, 1999*). We observed that *cat-2* mutant animals have disrupted pattern selectivity, as they showed reduced preference for pattern IV in the quad-chambers (*Figure 3A*). Addition of exogenous dopamine restored the accumulation of *cat-2* mutants within pattern IV (*Figure 3A*), demonstrating that the defect in spatial pattern selectivity of *cat-2* mutants is due to the lack of dopamine. To determine the dopamine receptor responsible for the selectivity, we analyzed mutant worms with impaired dopamine receptors. We found that disrupting the D2-like receptor DOP-3, but not other dopamine receptors (DOP-1, DOP-2, and DOP-4), led to a reduced preference for pattern IV (*Figure 3B*). DOP-3 is expressed in a variety of neurons in the head, ventral cord and tail (*Chase et al., 2004*). The site-of-action of DOP-3 for pattern selection is unknown.

To identify the molecular sensor that links mechanosensation, dopamine, and spatial pattern preference, we studied mutant worms that lack TRP-4, a stretch receptor expressed in the CEP, ADE,

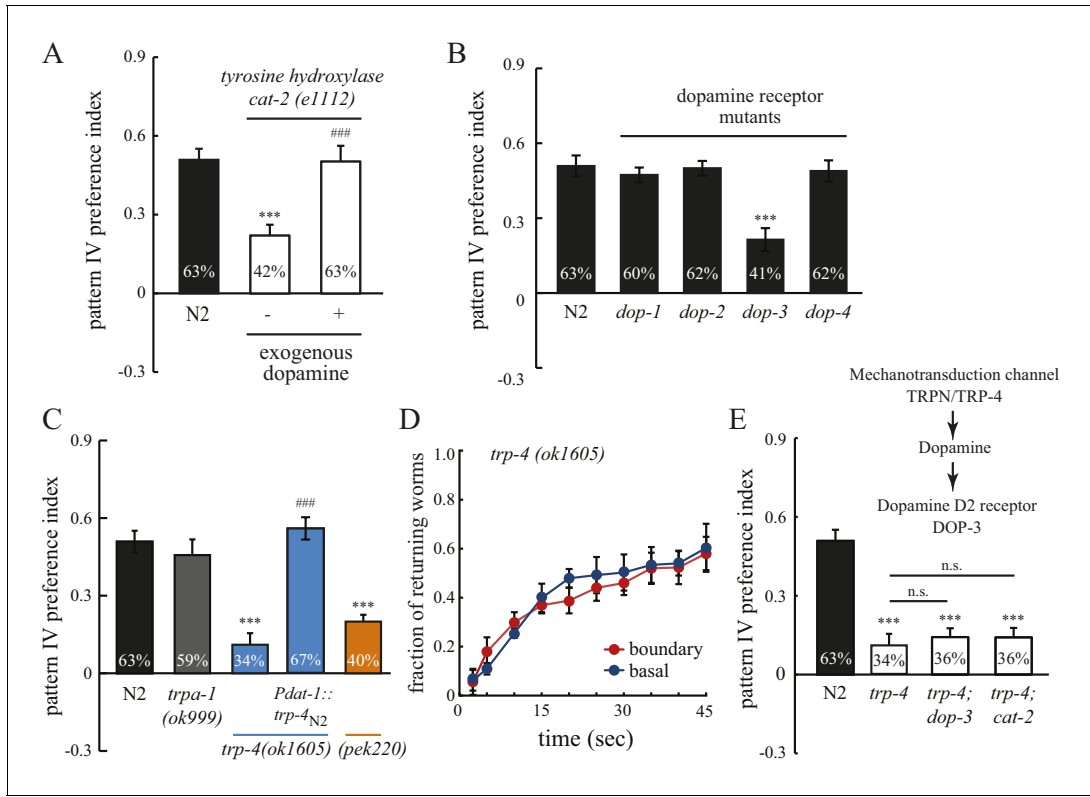

**Figure 3.** Spatial pattern selectivity of *C. elegans* requires dopamine signaling. (**A**) The requirement for dopamine biosynthesis. Mutant *cat-2* worms (*e1112*) that lack CAT-2 tyrosine hydroxylase show impaired preference for pattern IV, which is rescued by the addition of dopamine (2 mM). The number of independent repeats, n = 10; ***p<0.001 when compared with N2; ###p<0.001 when compared with *cat-2(e1112)* mutants; Tukey-Kramer multiple comparison tests. (**B**) The D2 dopamine receptor DOP-3 is required for spatial pattern selectivity. n = 12; ***p<0.001 when compared with N2; Dunnett's multiple comparison tests. (**C**) Preference for pattern IV requires the mechanotransduction TRP-4 channel in dopamine neurons. *trpa-1(ok999)* mutants were used as controls. *Pdat-1* was placed before the *trp-4* cDNA to drive specific expression in dopamine neurons. n = 12, ***p<0.001 when compared with N2, ***p<0.001 when compared with N2, ###p<0.001 when compared with *trp-4(ok1605)* mutants, Tukey-Kramer multiple comparison tests. (**D**) Mutant worms that lack TRP-4 do not show the enhanced returning behavior at boundaries. n = 6. (**E**) Double mutant worms (*trp-4; dop-3* and *trp-4; cat-2*) exhibited identical defects as the *trp-4* single mutant, suggesting that TRP-4, dopamine, and DOP-3 function in the same pathway. n = 12; ***p<0.001 when compared with N2; n.s.: not significant compared with *trp-4(ok1605)*; Tukey-Kramer multiple comparison tests. All data were collected 60 min after the worms were loaded into the chambers. Error bars denote s.e.m.

The following figure supplement is available for figure 3:

**Figure supplement 1.** The design strategy for a *trp-4* allele (*pek220*).

and PDE dopaminergic neurons and the DVA and DVC interneurons (*Hu et al., 2011*; *Kang et al., 2010*; *Li et al., 2006*). TRP-4 is the pore-forming subunit of mechanotransduction channels (*Kang et al., 2010*; *Li et al., 2006*). Mutant *trp-4(ok1605)* worms significantly reduced their preference for pattern IV (*Figure 3C*) and did not show the enhanced returning behavior after crossing the boundary between section IV and III (*Figure 3D*). To confirm that the defects in pattern selection are due to the lack of TRP-4 function, we generated a second allele *trp-4(pek220)* using CRISPR/CAS9 and homologous recombination (*Figure 3—figure supplement 1*). We found that *trp-4(pek220)* mutant worms exhibit similar defects as *trp-4(ok1605)* worms, confirming that the pattern IV preference requires TRP-4. The defects of *trp-4* single mutants were similar to *trp-4; cat-2* and *trp-4; dop-3* double mutants, indicating that *trp-4* acts in the same dopamine pathway as *cat-2* and *dop-3* (*Figure 3E*). Single-copy transgenes that express TRP-4 in dopaminergic neurons restored the preference for pattern IV (*Figure 3C*), further demonstrating a role of TRP-4 in dopaminergic neurons in mediating spatial pattern selectivity. Together, these data elucidate a signaling mechanism for the stretch receptor TRP-4, dopamine, and the dopamine receptor DOP-3, coupling spatial pattern detection with the behavioral preference of animals.

Recent studies have shown that natural genetic variations influence how *C. elegans* interacts with environmental cues such as food, oxygen, microbial pathogens, and neighboring worms (*Bendesky and Bargmann, 2011*; *Bendesky et al., 2012, 2011*; *Chang et al., 2011*; *Persson et al., 2009*). The physical structure of the environment is another important feature that varies with habitat. We therefore hypothesized that *C. elegans* strains from different natural habitats might differ in their selectivity of spatial patterns. We thus examined a wild isolate of *C. elegans*, Hawaiian CB4856 (designated HW) in the spatial pattern preference assay. Interestingly, we found that unlike N2, HW worms failed to aggregate in pattern IV of the quad-chambers (*Figure 4A, B*), and their returning behavior at boundaries was significantly reduced in comparison to N2 worms (*Figure 4C*). In addition, HW worms exhibited similar traveling speed and pause rates in uni- and quad-chambers (*Figure 4—figure supplement 1*), indicating that mechanosensory experience had little impact on HW behavior. Several studies have shown that behavioral differences between N2 and HW could be explained by laboratory-derived genetic changes (*e.g.,* allelic variations in *npr-1* and *glb-5*; reviewed in *Sterken et al. (2015)*. To test whether this is the case for spatial pattern selectivity, we examined the LSJ1 strain, a sibling of N2 that was cultivated separately after isolation. We found that the LSJ1 strain showed similar behavior to N2 worms in the spatial pattern preference assay (*Figure 4B*), suggesting that the difference in spatial pattern selectivity is not due to domestication-derived alleles.

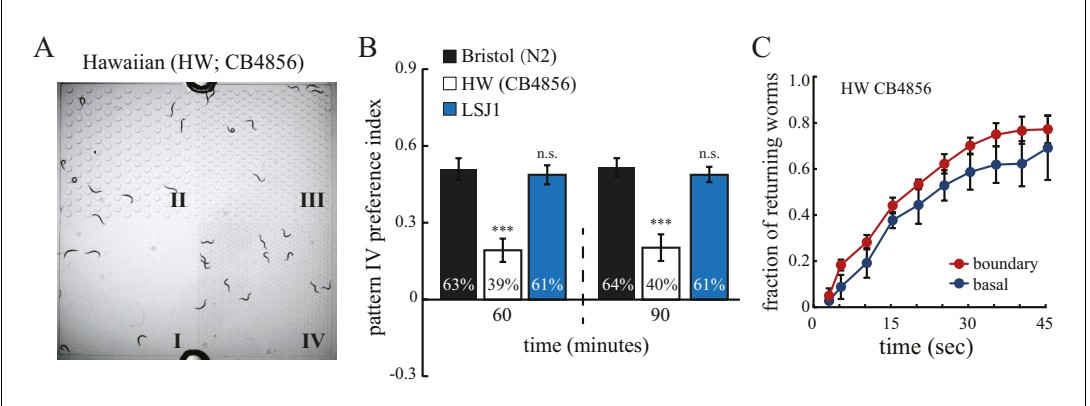

**Figure 4.** Hawaiian strain does not select pattern IV. (**A**) The Hawaiian CB4856 strain (HW) does not show significant preference for pattern IV of the quad-chamber. The representative image was taken at 60 min after the worms were loaded into the chamber. (**B**) Quantification of the preference for pattern IV of N2, HW, and LSJ1 (a sibling of the N2 strain) worms in the quad-chamber at 60 and 90 min. n = 12. (**C**) HW CB4856 worms do not show enhanced returning at boundaries. n = 6. Data were collected as described in *Figure 2B*.

The following figure supplement is available for figure 4:

**Figure supplement 1.** HW worms show identical locomotory behavior in quad- and uni- chambers.

Interestingly, the stretch receptor TRP-4 in the HW strain carries three amino acid substitutions (M36I, G41S, and E67G; *Figure 5A*) (*Thompson et al., 2013*). The *trp-4* gene is located on chromosome I in *C. elegans*. An N2/HW hybrid strain that carries the HW Chromosome I in an N2 background behaved like the HW strain in the spatial pattern preference assay. However, hybrid strains that carry other HW chromosomes in the N2 background were indistinguishable from N2 worms (*Figure 5B*), with the exception of those with HW Chromosome X, which showed a minor but significant reduction in the preference for pattern IV. These data are consistent with a role of *trp-4* in determining the natural variation in spatial pattern selectivity. They also suggest that other gene(s) on Chromosome X may contribute to the polymorphic behavior. Next, we isolated recombinant

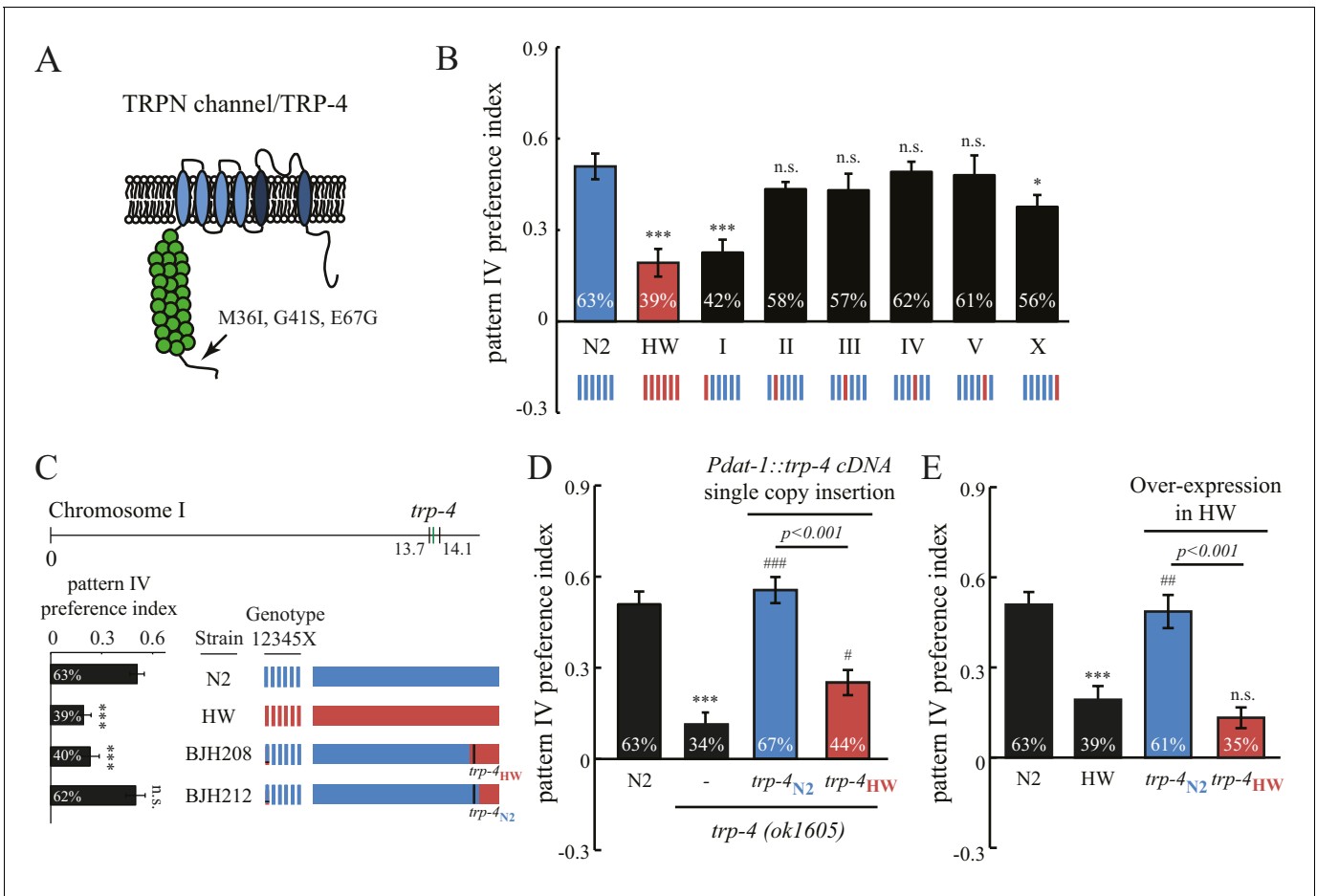

**Figure 5.** Polymorphisms in the mechanotransduction TRP-4 channel contribute to the variation of spatial pattern selectivity. (**A**) A cartoon illustrating the amino acid substitutions (M36I, G41S, and E67G) in HW TRP-4. (**B**) Chromosomes I and X were linked to altered spatial pattern selectivity between N2 and HW strains. Hybrid N2/HW strains carrying one HW chromosome (red) and five N2 chromosomes (blue) were quantified for spatial preference at 60 min. n = 12 for N2 and HW worms; n = 10 for worms with chromosome substitutions. (**C**) Recombinant inbred strains harboring N2 and HW *trp-4* genes showed distinct behavior in the spatial pattern preference test at 60 min. n = 12. Estimated regions of recombination (around 13.7 Mb in BJH208 and 14.1 Mb in BJH212) are indicated in the *upper panel*. The BJH208 strain that carries the N2 *trp-4* gene shows enrichment, but the BJH212 strain with HW *trp-4* does not (*lower panel*). (**D**) Transgenic *trp-4(ok1605)* worms expressing single-copy N2 or HW TRP-4 in dopaminergic neurons (*Pdat-1:: trp-4* cDNA) were tested in the spatial pattern preference assay. n = 12 for N2 and *trp-4(ok1605)* worms, n = 10 for worms with the transgenes. ***p<0.001, when compared to N2; ###p<0.001 and #p<0.05 when compared to *trp-4(ok1605) mutants*. Error bars denote s.e.m. (**E**) Transgenic HW worms carrying extrachromosomal arrays with either N2 or HW *trp-4* cDNA driven by *Pdat-1* were examined in the spatial pattern preference assay. n = 12. Data were collected 60 min after worm loading (unless specified otherwise). One-way ANOVA and Dunnett's multiple comparison tests were used for statistic analyses in *Figure 5*. ***p<0.001 when compared to N2; ##p<0.01 when compared to HW worms. Error bars denote s.e.m.

The following figure supplement is available for figure 5:

**Figure supplement 1.** Both N2 and HW *trp-4* cDNAs rescue defects in the swim-to-crawl transition of *trp-4(ok1605)* mutant worms.

inbred lines (RIL) from a cross between N2 and N2/HW hybrids that carry the HW Chromosome I. As shown in *Figure 5C*, two RIL strains behaved differently in the spatial pattern preference assay; one that carries the N2 *trp-4* gene showed preference for pattern IV, but another one with HW *trp-4* did not. To confirm the role of TRP-4 in spatial preference polymorphism, we performed rescue experiments using single-copy transgenes expressing either N2 or HW *trp-4* cDNA in dopaminergic neurons. N2 *trp-4* cDNA fully restored the selectivity defects of the *trp-4(ok1605)* mutant worms. By contrast, HW *trp-4* cDNA showed only low levels of rescue (*Figure 5D*). These data confirm that N2 TRP-4 is required for the spatial pattern preference observed in the Bristol strains. To test whether N2 TRP-4 is sufficient for endowing the HW strains with spatial pattern selectivity, we overexpressed N2 TRP-4 in HW worms. Because TRP channels are expected to form tetramers, we reasoned that overexpression of N2 TRP-4 might replace the HW TRP-4 channels. Indeed, we found that N2 TRP-4 but not HW TRP-4 overexpression caused HW worms to show spatial pattern selectivity similar to that of N2 (*Figure 5E*).

The substitution of three residues in HW TRP-4 occurs at the N-terminal region that is adjacent to Ankyrin repeats (*Figure 5A*). In flies, the Ankyrin repeats of NOMPC (the TRP-4 homologue) directly bind microtubules, and they are essential for mechanogating of the NOMPC channels (*Zhang et al., 2015*). To understand whether the three residue substitutions alter TRP-4 properties, we recorded electrophysiological responses of the dopaminergic CEP neurons to small mechanical displacements (*Figure 6A*) (*Kang et al., 2010*). The mechanoreceptor current in CEP neurons is mediated by TRP-4, and the amplitude of the current depends on the strength of the applied mechanical force. In agreement with a previous study (*Kang et al., 2010*), a displacement of 0.5 μm was sufficient to elicit TRP-4 mediated current in N2 worms, and the current was saturated at 3 μm displacement (*Figure 6B*). Interestingly, the sensitivity of HW TRP-4 to mechanical stimuli was significantly reduced. Larger mechanical displacement was required to generate currents with similar amplitudes (*Figure 6A–B*). These results are consistent with the findings that the N-terminal region of the TRP-4 homologue NOMPC is required for mechanogating in flies (*Zhang et al., 2015*). It is worth noting that, when mechanical stimuli are sufficiently strong, the HW TRP-4 is fully active (*Figure 6C*). Consistently, we found that an even higher density of pillars (novel pattern V) enhanced the enrichment of HW worms, but not of *trp-4(ok1605)* mutants (*Figure 6D*). Together, these findings indicate that amino acid substitutions in HW TRP-4 reduce the sensitivity to mechanical stimuli, which could subsequently alter spatial pattern perception in HW worms. Interestingly, whereas a single-copy HW TRP-4 transgene failed to restore spatial preference (*Figure 5E*), it completely rescued swimming/crawling transition in the *trp-4(ok1605)* mutant worms (*Figure 5—figure supplement 1*), suggesting that various types of TRP-4 dependent behaviors are tuned to different levels of mechanogating sensitivity.

Finally, we analyzed spatial pattern selectivity of ten additional wild isolates, for which complete genome sequences are available (*Thompson et al., 2013*). We found that five wild isolates exhibited a spatial preference like N2. Two isolates (AB1 and JU397) with substituted residues in TRP-4 exhibited defects in preference for pattern IV in the quad-chambers (*Figure 7A*). The TRP-4 channels in AB1 and JU397 each carry two substitutions (G41S and E67G), suggesting that the third substitution M36I in the HW strain is not required. A third isolate (CB4854) also carries two substitutions (E67G and I152M) in TRP-4; however, these worms did not show significant defects in the spatial pattern preference assay (*Figure 7A*). The most parsimonious explanation is that the G41S substitution is the key residue that modulates TRP-4 mechanosensitivity. However, it is also possible that E67G is important, but the combination of E67G and I152M suppresses the impact of E67G.

Intriguingly, three wild isolates (CB4853, JU300, and JU322) showed altered spatial pattern selectivity despite having N2-like TRP-4 sequence (*Figure 7A*). Closer examination of their genome sequences revealed that all three strains carry a substituted residue in DOP-3 at the same position (A273G). Upon editing the *dop-3* gene in N2 worms using CRISPR/Cas9 (*Friedland et al., 2013*), we found that the edited worms with the A273G substitution lost their spatial pattern selectivity (*Figure 7B*). These experiments indicate that the DOP-3 A273G mutation is sufficient to impair spatial pattern preference even with fully functional TRP-4. Therefore, the diversity of spatial pattern selectivity may arise from variations in either the detector that senses changes in the external world (*e.g.,* TRP-4), or the neuromodulator signals that define the internal state of animals (*e.g.,* dopamine and its receptors).

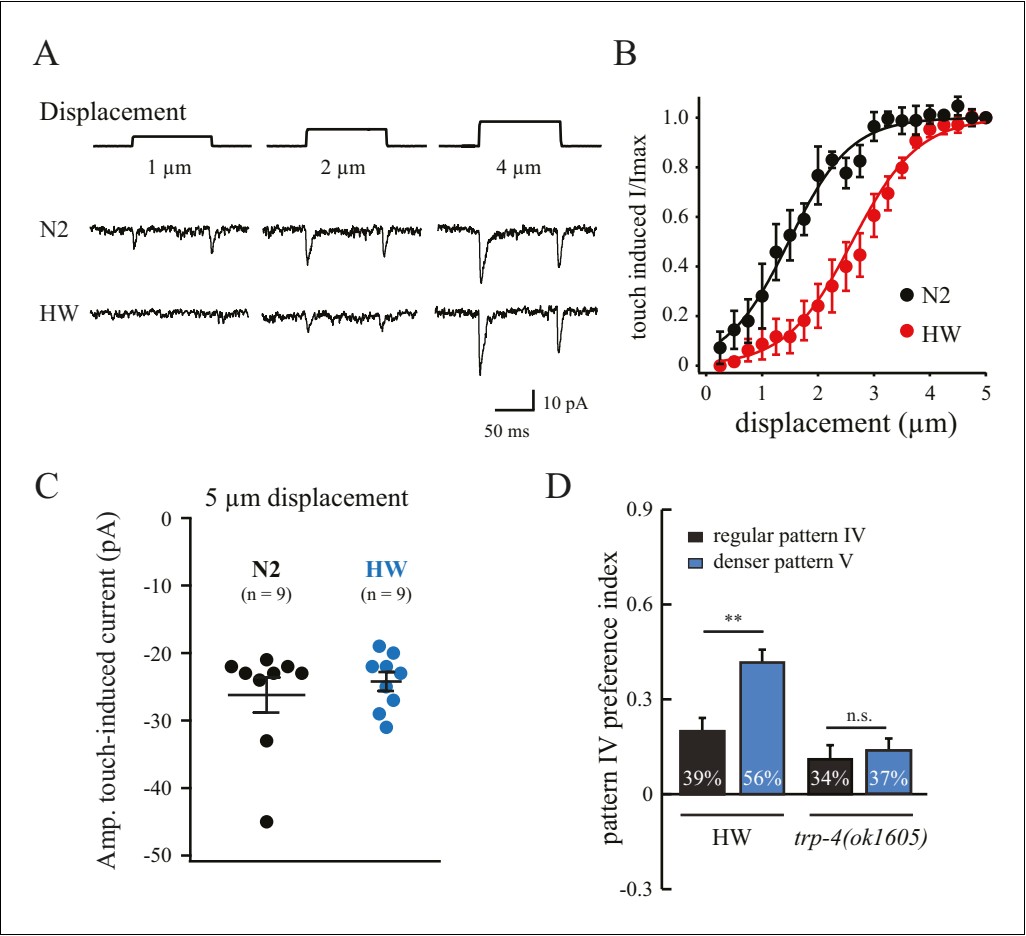

**Figure 6.** Polymorphisms in TRP-4 alter mechanosensitivity of dopaminergic neurons. (**A**) Representative traces of mechanoreceptor currents in wild type N2 and HW dopaminergic CEP neurons in response to small mechanical displacements (1 µm, 2 µm and 4 µm). Whole-cell patch clamp recordings (holding potential at −75 mV) were carried out at 20°C. (**B**) Stimulus-current curves of TRP-4 mechanogated on-currents in response to mechanical displacement. Data were fitted with a Boltzmann function. For N2 worms, the half-maximal displacement is 1.48 ± 0.04 µm and the slope factor is 0.56 ± 0.04 µm. For HW worms, the half-maximal displacement is 2.71 ± 0.03 µm, the slope factor is 0.58 ± 0.03 µm. n = 8 for N2 and n = 6 for HW; **p<0.01 and *p<0.05 when compared with HW worms. (**C**) A comparison of absolute current amplitudes in N2 and HW worms (at 5 µm displacements, n = 9 for N2 and n = 9 for HW). (**D**) Increases in pillar density in pattern IV enhanced HW aggregation but had little impact on *trp-4(ok1605)* mutant worms. Denser pattern V consisted of a pillar layout of 194 µm (pillar diameter) x 95 µm (distance between pillars). **p<0.01. Error bars denote s.e.m.

## Discussion

These results demonstrate that *C. elegans* is able to utilize touch sensation to discern different spatial patterns in the environment. Previous studies have shown that nematodes can evaluate their environment by sensing food (*Calhoun et al., 2015*) and chemicals (*Albrecht and Bargmann, 2011*). Our findings add a new dimension to the 'mind' of the worm by demonstrating a capability to perceive physical patterns in the environment. In many animal species, touch perception plays a key role in guiding exploration of the physical world. For example, insects carry antennae to collect tactile information during locomotion, rodents use whiskers to detect spatial settings, and sea mammals use facial hairs to trace changes in water flow patterns caused by swimming fish. Using a substantially simpler nervous system, *C. elegans* is able to detect spatial patterns, and to use mechanosensory information about the overall environment to form spatial pattern preferences. While primitive, this sense of physical space in worms is highly dynamic – it is tightly regulated by experience

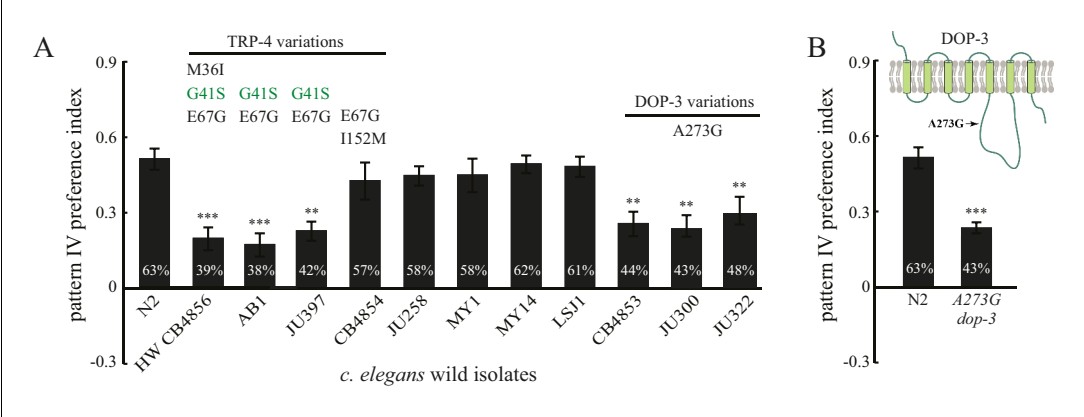

**Figure 7.** Natural variations in either TRP-4 or DOP-3 tune spatial pattern selectivity in *C. elegans*. (**A**) Spatial preference of *C. elegans* natural isolates was quantified using regular quad-chambers. n = 12 for each strain, ***p<0.001 and **p<0.01 when compared to N2; Dunnett's multiple comparison tests. (**B**) CRISPR/Cas9 edited N2 worms carrying the A273G substitution and wild type N2 worms were compared using the spatial pattern preference assay. n = 12; ***p<0.001, Student's *t* test. Data from the spatial pattern preference assay were collected at 60 min. Error bars denote s.e.m.

(*Figure 2E,F*), spatial context (*Figure 1—figure supplement 2D,E*), and food availability (*Figure 1—figure supplement 1C,D*). Together, our findings elucidate a basic but surprising sense of physical patterns in the surrounding environment in a simple nervous system of 302 neurons.

The selectivity of spatial patterns in *C. elegans* requires dopamine, which may serve as a 'reward' signal that connects the sensation of spatial patterns with innate preferences (*Schultz et al., 1997*). In higher animals, the existence of sensory modalities such as vision and hearing and associated brain structures, substantially expand the richness and quality of available information about the physical structure of their environment. Nevertheless, together with previous findings implicating dopamine in spatial attention (*Bellgrove et al., 2007*) and spatial memory (*Murphy et al., 1996*), as well as in disorders such as agoraphobia in humans (*Perna et al., 2011*), our study suggests that the coupling between dopamine signaling and a sense of spatial features is an ancestral feature of animal nervous systems.

## Materials and methods

### Nematode strains and growth

*C. elegans* strains were maintained under standard conditions at 20°C on nematode growth medium (NGM) agar plates (*Brenner, 1974*). These plates were seeded with *E. coli* OP50 lawns. The following strains were used in this study: (a) Wild type strains – AB1 (Adelaide, AU), CB4853 (Altadena, US), CB4854 (Altadena, USA), CB4856 (Hawaii, US), LSJ1 (Bristol, UK), JU258 (Madeira, PT), JU300 (Indre, FR), JU322 (Merlet, FR), JU397 (Hermanville, FR), MY1 (Emsland, DE), MY14 (Mecklenbeck, DE), and N2 (Bristol, UK); (b) Mutant strains – VC1141 *trp-4(ok1605)*, RB1052 *trpa-1(ok999)*, CB1112 *cat-2(e1112)*, LX645 *dop-1(vs100)*, LX702 *dop-2(vs105)*, LX703 *dop-3(vs106)*, FG58 *dop-4(tm1392)*, BJH2033 *trp-4(ok1605); dop-3(vs106)*, BJH2034 *trp-4(ok1605); cat-2(e1112)*; CX6448 *gcy-35(ok769)*; RB1115 *mec-10(ok1104)*; ZB2551 *mec-10(tm1552)*; (c) Chromosome substitution strains – WE5236 [I, CB4856 >N2], GN67 [II, CB4856 >N2], GN68 [III, CB4856 >N2], GN69 [IV, CB4856 >N2], GN70 [V, CB4856 >N2], GN231 [X, CB4856 >N2]; (d) Single-copy MosSCI strains – BJH189 *trp-4(ok1605); pekSi8 [cb-unc-119(+),Pdat-1::trp-4_{N2}]II*; BJH190 *trp-4(ok1605); pekSi9[cb-unc-119(+),Pdat-1::trp-4_{HW}]II*; BJH976 *mec-10(tm1552); pekSi102[NeoR(+),Pegl-44::mec-10]*; BJH977 *mec-10(tm1552); pekSi103[NeoR(+),Pmec-18::mec-10]*; BJH978 *mec-10(tm1552); pekSi104[NeoR(+),Pmec-10::mec-10]*; (e) Strains with engineered missense mutation by CRISPR-Cas9 – BJH701 *dop-3(pek201 [A273G])*; BJH720 *trp-4(pek220[GFP+hygR])*; (f) Strains with UV-integrated transgenes – BJH186 *pekIs85 [Pdat-1::trp-4_{N2}]* in CB4856, BJH187 *pekIs86 [Pdat-1::HW trp-4_{HW}]* in CB4856.

## Transgenes

Single copy transgenes were generated using the MosSCI technique (*Frøkjær-Jensen et al., 2012*; *Frøkjaer-Jensen et al., 2008*). Transgenes were inserted into the ttTi5605 Mos1 site on chromosome II. Multi-copy integrated transgenic strains for *trp-4* overexpression were generated by micro-injection of plasmids (BJP-B289: *Pdat-1::trp-4$_{N2}$* and BJP-B290: *Pdat-1::trp-4$_{HW}$*) at 50 ng μl$^{-1}$ into the HW CB4856 strain. The co-injection markers were *Pvha-6::mRuby* (BJP-B169) and *Pvha-6::GFP* (BJP-B197). Blank vector pBluescript was used as an injection filler to bring final DNA concentration to 100 ng μl$^{-1}$. Integration was obtained by UV irradiation of worms carrying the extrachromosomal arrays. Transgenic worms were outcrossed four times.

## CRISPR/CAS9

To construct the Cas9-sgRNA expression plasmid for making the *trp-4* deletion allele *pek220*, we replaced the gRNA sequence in the plasmid pJW1219 (Addgene; [*Ward, 2015*]) with the following sequence (5'-GTAAATGTCCCAGAACGCGG-3') using the overlapping PCR method. To construct the repair construct for homologous recombination, we inserted the left *trp-4* recombination arm, a disruption cassette containing *GFP::tbb-2utr + Prps-0::hygR::unc-54utr*, and the right *trp-4* recombination arm into pPD49.26 between SpeI and SalI and sites. The left *trp-4* recombination arm (~710 bp) was amplified using the forward primer 5'-CTTGCATGCCTGCAGGTCGACTTTGGAATTTCCTTTGATACAGCAACCCAC-3' and the reverse primer 5'-GTTCTTCTCCTTTACTGGAGGTGTGGGGATTCATCTGG-3'. The right *trp-4* recombination arm (~510 bp) was amplified using the forward primer 5'-GGCTCCTTGCGTTCATCTTCATGCACACGGCACATAGGACAGAG-3' and the reverse primer 5'-GGGCCCGTACGGCCGACTAGTCTTGGGAGTTTCCTCCCTTTTCC-3'.

## Fabrication of PDMS microfluidic chambers

Microfluidic chambers were fabricated using soft lithography with polydimethyl siloxane (PDMS, Sylgard 184, Dow Corning, MI). Silicon wafers were patterned with various designs using SU8-100 (Stanford microfluidic foundry, CA) and photolithography. PDMS was poured on top of the patterned silicon wafer, and was cured at 80°C for 2 hr. Cured PDMS was carefully lifted away from the silicon wafer. Sharpened stub needles (0.070 mm OD, 22G) were used to create the fluid inlet and outlet paths on the PDMS device. A plasma cleaner PDC-32G (Harrick Scientific, NY) was used to treat the PDMS device for 1 min. Glass slides (1 mm thick) were immediately bound to the devices after treating. Distilled water was added to the devices within 5 min. Before loading the worms, the devices were washed with an ample amount of M9 buffer.

## Behavior assay for spatial pattern preference

Worms were synchronized for the behavioral assays. Fifty gravid adult worms were transferred onto a 10 cm NGM plate seeded with OP50 to lay eggs for an hour. Then adult worms were removed by picking. Eggs were grown at 20°C for 72 hr to reach the young adult stage for behavioral tests. The PDMS chambers were washed with M9 buffer. Synchronized worms (72 hr) were transferred using glass Pasteur pipettes into Eppendorf tubes. After settling by gravity, worms were washed once to remove residual bacteria OP50. Approximately 30–70 worms were loaded into the inlet, and were allowed to freely explore the chamber. The location of the worms was recorded for 10 s using a Moticam 2500 CCD camera mounted on a stereo microscope (Leica, Germany) at various time points as indicated in the study.

Worm preference for each pattern was quantified as the percentage of worms occupying that pattern in the quad-chambers and, in addition, using a spatial pattern preference index. The index was calculated for each pattern using the following formula: [# of worms in one pattern – (# of worms in the other three patterns/3)] ÷ total # of worms. An index value of 0 indicates in all patterns equal distribution among the different patterns. An index of −0.33 for a certain pattern indicates no worms found in that pattern. An index of 1 for a certain pattern indicates that all worms are present in that pattern. Each experiment was replicated 3–5 times. 'n' indicates the number of independent repeats of a particular experiment. Results were analyzed in a blind manner. To analyze the speed of locomotion and the pause rate, worm behavior in quad-chambers (between 58 to 61 min) was recorded using a Moticam 2500 digital camera that was mounted on a Leica S6D microscope. Videos were processed offline using MATLAB scripts developed by *Albrecht and Bargmann (2011)*. The

animal's locomotory state (forward, reverse, and turn) at boundaries was determined by visual inspection of the video.

## Statistical analysis

The following statistic methods were used to analyze behavioral data. For single pairwise comparisons, we used the Student's $t$ test. For analyses that require comparisons between multiple means to a control mean, we used one-way ANOVA followed by Dunnett's multiple comparison tests. For analyses that require comparisons among all possible pairs of means, we used the Tukey-Kramer method. Statistical analysis was performed using Igor Pro 6. Mean and standard error of the mean (SEM) are reported. Results were considered statistically significant when the p-value was <0.05.

## Electrophysiological recording

Young adult worms were immobilized on sylgard-coated coverslips using a cyanoacrylate glue Histo-acryl Blue (Aesculap, PA). Worm dissection was performed in extracellular solution (*Dong et al., 2015*). To expose the cell body of the CEP neuron, a small piece of the worm cuticle was cut open as previously described (*Kang et al., 2010*). Dissected animals were mounted onto a fixed stage upright fluorescence BX51WI microscope (Olympus, Japan) equipped with a 60x/1.0NA water immersion objective lens, Nomarski-DIC optics, an X-cite 200DC light source (Excelitas, MA), and a ORCA-Flash2.8 CMOS camera (Hamamatsu Corp., Japan). All worms carried a Pdat-1::GFP transgene (*Lints and Emmons, 1999*) enabling CEP identification by green fluorescence.

Recording pipettes were pulled from borosilicate glass capillaries using a P-97 micropipette puller (Sutter Instruments, CA). These pipettes had resistances of ~15 MΩ when filled with internal solution (135 CH3O3SCs, 5 CsCl, 5 MgCl2, 5 EGTA, 0.25 CaCl2, 10 HEPES and 5 Na2ATP, adjusted to pH 7.2 using CsOH). Extracellular solution contained (in mM) 150 NaCl, 5 KCl, 1 CaCl2, 5 MgCl2, 10 glucose and 10 HEPES, titrated to pH 7.3 with NaOH, 330 mOsm with sucrose. All chemicals were purchased from Sigma.

Whole-cell patch clamp recordings (holding potential at −75 mV) were carried out by using HEKA EPC-10 double amplifier controlled by HEKA Patchmaster software at 20°C. The junction potential was measured by ±10 mv 5 ms square testing signals. After the pipette tip was inserted into the bath solution, the junction potential was corrected using the 'Auto V0' function, which calls a procedure for automatic zeroing of the pipette current. Just before the pipette touched the cell membrane for the seal, the 'Auto V0' was used again to make sure the junction potential was well corrected. The currents were amplified and acquired using an EPC-10 amplifier (HEKA, Germany). The signal was digitized at 20 kHz and filtered at 2 kHz using Patchmaster (HEKA, Germany). A glass stimulus probe was mounted to a piezo-driven actuator (Physik Instrumente, Germany) held by a micromanipulator. Displacement was triggered and controlled by the piezo amplifier E625 (PI), which is synchronized with programmed signals from HEKA Patchmaster software. A voltage step of 222 mV was used to trigger each 1 µm displacement of the probe. Series resistance was compensated to 70%. Electrophysiological data were analyzed using Igor Pro 6 (Wavemetrics, OR).

## Acknowledgements

We thank Drs. Michael Ailion, Harmit Malik, Irini Topalidou, and members of the Bai lab for comments. We thank the *C. elegans* Genetics Stock Center, Addgene, Dr. William Schafer, Dr. Irini Topalidou, and Dr. Erik Andersen for strains and reagents. We thank Ms. Shangjun Zhao for technical assistance. This work was supported by funds from the National Institutes of Health (R01-NS085214), Hartwell Innovation Fund, and FHCRC New Development Grant to JB.

## Additional information

### Funding

| Funder | Grant reference number | Author |
| --- | --- | --- |
| NIH Office of the Director | R01-NS085214 | Jihong Bai |
| Hartwell Foundation | 2015 Hartwell Innovation | Jihong Bai |

Pilot Funds

The funders had no role in study design, data collection and interpretation, or the decision to submit the work for publication.

## Author contributions

BH, Conceptualization, Data curation, Formal analysis, Investigation, Writing—original draft, Designed and carried out experiments, collected the data and assisted with preparing the manuscript; YD, Data curation, Validation, Investigation, Designed and carried out electrophysiology experiments, Collected and analyzed the data; LZ, Validation, Methodology, Designed and generated transgenic C. elegans strains, Assisted with behavioral testing and validation; YL, Assisted with behavioral testing, data collection, data analysis, and validation; IR, Writing—original draft, Writing—review and editing, Assisted with preparing and revising the manuscript; JB, Conceptualization, Supervision, Funding acquisition, Investigation, Methodology, Writing—original draft, Writing—review and editing, Prepare and revise the manuscript, designed all experiments, and supervised the project

## Author ORCIDs

Yongming Dong, http://orcid.org/0000-0002-6510-0913

Lin Zhang, http://orcid.org/0000-0002-9065-4168

Yan Liu, http://orcid.org/0000-0002-5631-7272

Ithai Rabinowitch, http://orcid.org/0000-0003-0361-9055

Jihong Bai, http://orcid.org/0000-0001-6773-2175

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
