## [Decision Letter]

Thank you for submitting your article "Dopamine Signaling Tunes Spatial Pattern Selectivity in *C. elegans*" for consideration by *eLife*. Your article has been reviewed by two peer reviewers, and the evaluation has been overseen by a Reviewing Editor and Eve Marder as the Senior Editor. The reviewers have opted to remain anonymous.

The comments of the two reviewers can be found below. You will see that the reviewers have convergent comments that I request you to address. In essence these are the two sets of key experiments that we would want to see in a revised version:

1) Validate their results with (null) alleles of *mec-10* and *trp-4*.

2) Rescue of the *mec-10* phenotype to determine its currently very unclear focus of action (determination of focus of action of *dop-3* does not need to be performed, but possible site of action in VNC MNs should be discussed).

*Reviewer #1:*

In this paper, the authors demonstrate a role for dopaminergic neurons in *C. elegans* in a behavior though which worms preferentially accumulate in environments with particular mechanosensory characteristics. Specifically, they use specially fabricated microfluidic environments to show that worms prefer regions with a high density of mechanical obstacles, and that this behavior requires the mechanosensory *trp* channel *trp-4* and the dopamine receptor *dop-3*. They also show that natural variation in these two genes contributes to variability in the observed behavior; in the case of *trp-4*, they go on to show that the genetic variants have altered sensitivity of *trp-4*-mediated mechanotransdution.

Overall, this is an interesting story that brings together a variety of experimental threads to provide a convincingly simple and straightforward explanation of a seemingly complex behavior. I do think there are a few small missing pieces which if added to the study would make the molecular and cellular explanation of the behavior significantly more complete.

1) In the manuscript, the authors state that "worms sense localchanges in pattern density, and respond quickly to unfavorable changes by reorientation," (Results). However, I don't think the authors present any direct evidence that reorientation (i.e. turning/reversal rates) change when crossing from higher to lower obstacle density. In principle, could the accumulation in more dense environments simply involve an orthokinesis mechanism, by which locomotion rates are higher in low-density and lower in high-density environments? This seems particularly relevant as dopamine (and *dop-3*) are known to control speed but not to my knowledge reorientation. It would be good for the authors to directly test turning/reversal rates (with a correction for the known differences in speed).

2) Although there is a significant amount of data in the paper about the role of *dop-3*, they surprisingly never do cell-specific rescue experiments to see where the DOP-3 receptor is acting. The previous work of Chase et al. suggests it could function in ventral cord motorneurons, and it would be straightforward to see if expression there is sufficient to rescue their phenotypes. This could fill in a missing gap to provide a satisfying molecular/circuit model for the behavior.

3) The authors show that *mec-10* mutants are defective in the behavior and cite this as evidence that mechanosensation is involved. But MEC-10 is not known to play a direct role in mechanosensation by the dopamine neurons and indeed is not even expressed in those cells. The only plausible explanation I can think of is that since *mec-10* functions in the FLP neurons, and the FLPs and CEPs are known to modulate each other's activity through a gap junction circuit of nose touch neurons (Chatzigeorgiou 2011, Rabinowitch 2013), *mec-10* could influence CEP activity indirectly through this circuit. They could test this easily by seeing if the *mec-10* phenotype rescues under an FLP promoter. As with point 2, this could tie what now seems like a random observation into a convincing circuit model.

*Reviewer #2:*

The manuscript from Han et al. evaluates the ability of *C. elegans* nematodes to use surface texture as a navigational aid (a process called thigmotaxis in other literature). The study introduces a behavioral assay for texture preference based on an arena fabricated using soft-lithography techniques and evaluates the effect of mutations in two ion channel genes-*mec-10* and *trp-4*. Additionally, the work links mechanosensation to dopaminergic signaling and its reception by the DOP-3 receptor. The *mec-10* gene encodes a pore formingsubunit of the sensory mechanotransduction channel in the gentle touch receptor neurons (TRNs) and the *trp-4* gene encodes a pore-forming subunit of the sensory mechanotransduction channel expressed by texture-sensing and proprioceptive neurons (CEP, ADE, PDE). The TRP-4 protein is homologous to *Drosophila* NOMPC and is predicted to form a non-selective cation channel activated by mechanical force. The main findings are that *C. elegans* nematodes perform thigmotaxis, that this behavior differs among wild isolates and depends on TRP-4 ion channels as well as dopaminergic signaling.

The data are presented in a clear and engaging manner, certain findings rely on mutant alleles that do not cleanly or clearly eliminate the cellular or molecular function in question (see below). For this reason, this reviewer believes that additional experimental work is essential to support the current conclusions.

1) The role of *mec-10*-expressing mechanoreceptor neurons in quadrant assays (part 1)

The preference defect in *mec-10(ok1104*) mutants is as strong as that found for *trp-4(ok1605*) mutants (compare Figure 1—figure supplement 2 with Figure 5). Thus, it is puzzling that the authors report the *mec-10* result as supplementary material and even more puzzling that they have chosen to focus exclusively on *trp-4*. Given that these genes are expressed in non-overlapping sets of mechanoreceptor neurons, it seems essential to understand determine whether or not these genes act together to mediate the observed thigmotaxis behaviors.

2) The role of *mec-10*-expressing mechanoreceptor neurons in quadrant assays (part 2)

The authors study *mec-10(ok1104*) and find that these animals have altered responses to pillar arrays. This conclusion is predicated on the notion that the *ok1104* mutation disables mechanosensation – (Results): "mutant worms with dysfunctional MEC-10 DEG/ENaC mechanosensory channels showed a significant reduction in their preference for pattern IV (Figure 1—figure supplement 2), demonstrating that spatial pattern selectivity requires proper mechanosensation." But, the situation is much more complex. First, *mec-10* is expressed in three classes of mechanoreceptor neurons, the touch receptor neurons (TRN), PVD and FLP. Which neurons are responsible for this effect? Second, the *ok1104* allele has very mild effects on TRN-mediated mechanosensation. In fact, Arnadottir et al. (J Neurosci, 2011) showed that *ok1104* mutants retain nearly normal behavioral responses to touch and mechanoreceptor currents. Additionally, the *ok1104* allele deletes 143bp of the gene and allows for the transcription of at least a single transcript encoding an abberant protein. Thus, the *ok1104* allele is unlikely to be null. Thus, the observed defects in behavioral assays could be due to the expression of a defective protein in the TRNs, PVD, or FLP. At a minimum, the authors need to communicate this complexity and uncertainty to readers. It would be better, however, to evaluate additional alleles, including *tm1552*, a different deletion in *mec-10* and *u253* a deletion in *mec-4* that eliminates mechanosensory currents in the TRNs and behavioral sensitivity.

3) The contribution of *trp-4* to responses to pillar assays

There are three alleles encoding deletions in the *trp-4* gene available from the CGC stock center: *gk341, sy695*, and *ok1605*. It is unfortunate that the authors have chosen *ok1605* for this study-this allele encodes an in-frame deletion that may not be null. Thus, the behavioral deficit observed could be the result of an aberrant TRP-4 protein rather than the absence of TRP-4. The interpretation of the authors' examination of *trp-4* and its genetic polymorphisms depends on resolving this uncertainty.

4) Differences in current-displacement curves in N2 and HW

The recordings in Figure 6 are presented in an ambiguous manner and critical information about the recording conditions are missing.

Did the authors record from CEP neurons in wild-type N2 and HW worms? Or, were recordings performed in transgenic N2 animals expressing TRP-4(N2) and TRP-4(HW) from transgenes?

If the latter, were the recordings performed in a wild-type N2 background or in *trp-4(ok1605*)? What holding potential used to record currents? Please add this information the figure and/or its legend.

Where voltages were corrected for junction potentials?

Do the maximal currents differ between N2 and HW? Please show pooled data for absolute current amplitude for 5µm displacements that saturate the current-displacement curves in both cases.

The answers to these questions affect interpretation, especially in light of point#3 above. The authors ascribe the difference displacement sensitivity to polymorphisms in the *trp-4* gene. While this inference is consistent with the available data, it is not the only possible interpretation. For example, if the recordings were in N2 and HW genetic backgrounds-variations in accessory proteins could just as easily account for the observed effects.

5) Variation in behavioral preferences among wild isolates

The difference between N2 and HW is striking and the authors' efforts to understand this source of this phenotypic difference are a terrific first start. However, their inferences about the causal role played by SNPs in the *trp-4* gene extend beyond the data available. For instance, since *trp-4* is not the only single gene that can mutate to alter this behavior (see point #1 above and the defect that appears to be linked to the X chromosome), the exclusive focus on *trp-4* is not really warranted. In other words, the phenotypic differences between wild isolates may be a combined function of differences in *trp-4* function and other factors.

6) Behavioral mechanism of preferences in quadrant assay

Throughout the description of the quadrant assay, this reviewer was struck by the concept of preference. Given the authors' finding that animals move more slowly in the denser patterns, isn't this arena essentially a speed trap? (i.e. animals accumulate in pattern IV because once they enter this zone, they move more slowly. The fact that this preference declines in longer assays – Figure 1—figure supplement 1 – supports this alternative interpretation). It would be useful to distinguish between the apparent preference and how this preference is produced.

---

## [Author Response]

*[…] Reviewer #1:*

*[…] 1) In the manuscript, the authors state that "worms sense localchanges in pattern density, and respond quickly to unfavorable changes by reorientation," (Results). However, I don't think the authors present any direct evidence that reorientation (i.e. turning/reversal rates) change when crossing from higher to lower obstacle density. In principle, could the accumulation in more dense environments simply involve an orthokinesis mechanism, by which locomotion rates are higher in low-density and lower in high-density environments? This seems particularly relevant as dopamine (and dop-3) are known to control speed but not to my knowledge reorientation. It would be good for the authors to directly test turning/reversal rates (with a correction for the known differences in speed).*

In the revised manuscript, we have added new results showing that worms reorient at the boundaries (increased turns and reversals; Figure 2). Dopamine has been shown to increase the frequency of high-angled turns during area-restricted search behavior (Hill et al. 2004). Consistent with a role of DOP-3 in reorientation, a previous study suggests that *dop-3* mutant animals make longer-lasting turns, which result in curvier trajectories (Stephens et al., 2010).

*2) Although there is a significant amount of data in the paper about the role of dop-3, they surprisingly never do cell-specific rescue experiments to see where the DOP-3 receptor is acting. The previous work of Chase et al. suggests it could function in ventral cord motorneurons, and it would be straightforward to see if expression there is sufficient to rescue their phenotypes. This could fill in a missing gap to provide a satisfying molecular/circuit model for the behavior.*

DOP-3 is strongly expressed in GABAergic motoneurons, as well as many neurons in the head and the tail. To ask if DOP-3 functions in GABAergic neurons, we have expressed single-copy *Punc-47::dop-3* transgenes in *dop-3(vs106)* mutant worms. However, DOP-3 in GABAergic neurons fails to rescue the defects, indicating that DOP-3 expression in GABAergic motoneurons is not sufficient to support spatial pattern selection. Because these negative results do not demonstrate the site-of-action of DOP-3, we have not included them in the manuscript. However, if the reviewer feels that these data are essential, we can add these results into the manuscript.

The broad expression pattern of DOP-3 has made it difficult to precisely identify the site-of action. We agree with the reviewer that cell-specific experiments will fill in the missing knowledge gap. We continue to perform these experiments, but it will require much more effort before knowing the answer.

*3) The authors show that mec-10 mutants are defective in the behavior and cite this as evidence that mechanosensation is involved. But MEC-10 is not known to play a direct role in mechanosensation by the dopamine neurons and indeed is not even expressed in those cells. The only plausible explanation I can think of is that since mec-10 functions in the FLP neurons, and the FLPs and CEPs are known to modulate each other's activity through a gap junction circuit of nose touch neurons (Chatzigeorgiou 2011, Rabinowitch 2013), mec-10 could influence CEP activity indirectly through this circuit. They could test this easily by seeing if the mec-10 phenotype rescues under an FLP promoter. As with point 2, this could tie what now seems like a random observation into a convincing circuit model.*

We thank the reviewer for constructive insights. To understand where MEC-10 functions, we have carried out cell-specific rescue experiments (Figure 1). We find that single-copy *Pegl-44::mec-10* transgenes fully rescue the selection defects in *mec-10(tm1552)* mutant worms. By contrast, single-copy *Pmec-18::mec-10* transgenes fail to rescue. These results confirm the reviewer’s prediction that MEC-10 functions in FLPs but not in TRNs.

The text has been modified as follows: “By contrast, mutant worms with dysfunctional MEC-10 DEG/ENaC channels showed a significant reduction in their preference for pattern IV (Figure 1). […] Together, these data indicate that MEC-10 acts in the polymodal nociceptive FLP neurons to support the selection of pattern IV.”

Reviewer #2:

*[…] 1) The role of mec-10-expressing mechanoreceptor neurons in quadrant assays (part 1)*

*The preference defect in mec-10(ok1104) mutants is as strong as that found for trp-4(ok1605) mutants (compare Figure 1—figure supplement 2 with Figure 5). Thus, it is puzzling that the authors report the mec-10 result as supplementary material and even more puzzling that they have chosen to focus exclusively on trp-4. Given that these genes are expressed in non-overlapping sets of mechanoreceptor neurons, it seems essential to understand determine whether or not these genes act together to mediate the observed thigmotaxis behaviors.*

The main reason for our focus on TRP-4 is due to the tight link between Chromosome 1 and the changes in pattern IV preference. While we have also observed that chromosome X is linked to behavior changes, the small alteration in the hybrid strain (Figure 5) has made it very difficult to further identify the linkage region. Therefore, we have focused on the traceable linkage in the *trp-4* region on Chromosome 1. We agree with the reviewer that *trp-4* is not the only change between N2 and CB4856 strains. Indeed, we have stated in the previous manuscript that “other gene(s) on Chromosome X may contribute to the polymorphic behavior”.

We have carried out several experiments to further investigate the role of MEC-10. First, we have studied another *mec-10* allele *tm1552*. Our results confirm that disruption of MEC-10 reduces the preference for pattern IV. Second, cell-specific rescue experiments show that MEC-10 functions in FLP neurons (please see response to point 3 by reviewer 1). These data are now presented in Figure 1 in the revised manuscript.

*2) The role of mec-10-expressing mechanoreceptor neurons in quadrant assays (part 2)*

*The authors study mec-10(ok1104) and find that these animals have altered responses to pillar arrays. This conclusion is predicated on the notion that the ok1104 mutation disables mechanosensation – (Results): "mutant worms with dysfunctional MEC-10 DEG/ENaC mechanosensory channels showed a significant reduction in their preference for pattern IV (Figure 1—figure supplement 2), demonstrating that spatial pattern selectivity requires proper mechanosensation." But, the situation is much more complex. First, mec-10 is expressed in three classes of mechanoreceptor neurons, the touch receptor neurons (TRN), PVD and FLP. Which neurons are responsible for this effect?*

In the revised manuscript, we include a set of results from cell-specific rescue experiments (Figure 1). As we discussed above, our results show that MEC-10 functions in FLPs.

*Second, the ok1104 allele has very mild effects on TRN-mediated mechanosensation. In fact, Arnadottir et al. (J Neurosci, 2011) showed that ok1104 mutants retain nearly normal behavioral responses to touch and mechanoreceptor currents. Additionally, the ok1104 allele deletes 143bp of the gene and allows for the transcription of at least a single transcript encoding an abberant protein. Thus, the ok1104 allele is unlikely to be null. Thus, the observed defects in behavioral assays could be due to the expression of a defective protein in the TRNs, PVD, or FLP. At a minimum, the authors need to communicate this complexity and uncertainty to readers. It would be better, however, to evaluate additional alleles, including tm1552, a different deletion in mec-10 and u253 a deletion in mec-4 that eliminates mechanosensory currents in the TRNs and behavioral sensitivity.*

We have evaluated the *mec-10(tm1552)* allele, as suggested by the reviewer. We show that the preference for pattern IV is disrupted in this putative null allele (Figure 1).

*3) The contribution of trp-4 to responses to pillar assays*

*There are three alleles encoding deletions in the trp-4 gene available from the CGC stock center: gk341, sy695, and ok1605. It is unfortunate that the authors have chosen ok1605 for this study-this allele encodes an in-frame deletion that may not be null. Thus, the behavioral deficit observed could be the result of an aberrant TRP-4 protein rather than the absence of TRP-4. The interpretation of the authors' examination of trp-4 and its genetic polymorphisms depends on resolving this uncertainty.*

To address this question, we have generated a new *trp-4* allele (*pek220)* using CRISPR/CAS9 and homologous recombination methods. As shown in Figure 3—figure supplement 1 disruption cassette that contains *gfp::tbb-2utr* and *Prps-0::hygR::unc-54utr* was inserted into *trp-4* exon 2 (Figure 3—figure supplement 1). Insertion of this disruption cassette results in a deletion and an early stop codon that put *trp-4* out-of-frame after the amino acid residue 53. The mutant *trp-4(pek220)* worms exhibit similar defects in the behavioral test as *trp-4(ok1605)* worms (Figure 3), confirming that pattern IV preference requires TRP-4. Furthermore, we have shown that the behavioral defect in *trp-4*(*ok1605)* worms can be fully rescued by *trp-4* cDNA expression in dopaminergic neurons. Therefore, the behavioral defects of *trp-4(ok1605)* worms are due to the loss of TRP-4 function.

*4) Differences in current-displacement curves in N2 and HW*

*The recordings in Figure 6 are presented in an ambiguous manner and critical information about the recording conditions are missing.*

We have modified the text to clarify the recording conditions, as detailed below.

*Did the authors record from CEP neurons in wild-type N2 and HW worms? Or, were recordings performed in transgenic N2 animals expressing TRP-4(N2) and TRP-4(HW) from transgenes?*

*If the latter, were the recordings performed in a wild-type N2 background or in trp-4(ok1605)?*

The recordings were performed using wild type N2 and HW strains. We have clarified this point by stating “Representative traces of mechanoreceptor currents in wild type N2 and HW dopaminergic CEP neurons …” in the legend for Figure 6.

What holding potential used to record currents? Please add this information the figure and/or its legend.

We have added the following statement “Whole-cell patch clamp recordings (holding potential at −75 mV) were carried out at 20°C” in the figure legend.

*Where voltages were corrected for junction potentials?*

The following text was added to the Methods section: “The junction potential was measured by ± 10mv 5ms square testing signals. After the pipette tip was inserted into the bath solution, the junction potential was corrected using the “Auto V0” function, which calls a procedure for automatic zeroing of the pipette current. Just before the pipette touched the cell membrane for seal, the “Auto V0” was used again to make sure the junction potential was well corrected.”

*Do the maximal currents differ between N2 and HW? Please show pooled data for absolute current amplitude for 5µm displacements that saturate the current-displacement curves in both cases.*

*The answers to these questions affect interpretation, especially in light of point #3 above. The authors ascribe the difference displacement sensitivity to polymorphisms in the trp-4 gene. While this inference is consistent with the available data, it is not the only possible interpretation. For example, if the recordings were in N2 and HW genetic backgrounds-variations in accessory proteins could just as easily account for the observed effects.*

As suggested by the reviewer, we have included a comparison of absolute current amplitudes in N2 and HW worms in Figure 6. No significant difference was detected, which suggests that the HW TRP-4 is fully active when mechanical stimuli are sufficiently strong.

Because results from electrophysiological recordings could be explained by changes in accessory proteins, we also assayed N2 and HW cDNAs for their ability to rescue behavioral defects. We show that N2 and HW *trp-4* have a significant difference in their ability to rescue the behavioral defects in *trp-4(ok1605)* mutant worms (Figure 5). We also show that transgenic HW worms that carry N2 trp-4 exhibit increased preference of pattern IV when compared to those HW worms expressing HW trp-4 (Figure 5). These results are unlikely due to changes in accessory proteins, as worms with the same genetic background were used for comparison.

*5) Variation in behavioral preferences among wild isolates*

*The difference between N2 and HW is striking and the authors' efforts to understand this source of this phenotypic difference are a terrific first start. However, their inferences about the causal role played by SNPs in the trp-4 gene extend beyond the data available. For instance, since trp-4 is not the only single gene that can mutate to alter this behavior (see point #1 above and the defect that appears to be linked to the X chromosome), the exclusive focus on trp-4 is not really warranted. In other words, the phenotypic differences between wild isolates may be a combined function of differences in trp-4 function and other factors.*

As discussed above, we agree that *trp-4* is unlikely to be the only single gene accounting for the phenotypic differences among wild isolates. First, we have stated that, “other gene(s) on Chromosome X may contribute to the polymorphic behavior”. Second, we have shown that an additional SNP in DOP-3 is likely to cause behavioral changes in some wild isolates (Figure 7).

The main reason of our TRP-4 focus is the tight link between chromosome 1 and the reduced preference for pattern IV. This link is traceable and we found that it ties to *trp-4* in the case of the HW strain. In general, we agree with the reviewer that many factors may contribute to the phenotypic differences. What our data have shown is that SNPs in *trp-4* and *dop-3* produce traceable and significant changes.

*6) Behavioral mechanism of preferences in quadrant assay*

*Throughout the description of the quadrant assay, this reviewer was struck by the concept of preference. Given the authors' finding that animals move more slowly in the denser patterns, isn't this arena essentially a speed trap? (i.e. animals accumulate in pattern IV because once they enter this zone, they move more slowly. The fact that this preference declines in longer assays – Figure 1—figure supplement 1 – supports this alternative interpretation). It would be useful to distinguish between the apparent preference and how this preference is produced.*

We find that behavioral mechanisms in the quadrant assay are complex and context-dependent. Both reversal and turning rates increase once worms step out of the pattern IV (Figure 2), suggesting that additional mechanisms besides locomotion speed are involved. Interestingly, the reduction in locomotion speed in pattern IV is context-dependent, as we find that neighboring patterns have a significant impact on worm locomotion (Figure 2). Furthermore, the selection for pattern IV depends on feeding state, suggesting that this behavior is not fixed to the physical design of the patterns. Based on these observations, we propose that several behavioral strategies including changes in speed and turning are engaged to produce the plastic behavior of pattern selection.